# Increased Intrahepatic Mast Cell Density in Liver Cirrhosis Due to MASLD and Other Non-Infectious Chronic Liver Diseases

**DOI:** 10.3390/ijms27010392

**Published:** 2025-12-30

**Authors:** Nicolás Ortiz-López, Araceli Pinto-León, Javiera Favi, Dannette Guíñez Francois, Larissa Aleman, Laura Carreño-Toro, Alejandra Zazueta, Fabien Magne, Jaime Poniachik, Caroll J. Beltrán

**Affiliations:** 1Laboratory of Immunogastroenterology I, Section of Gastroenterology, Department of Medicine, Hospital Clínico Universidad de Chile, Santiago 8380456, Chile; nicolas.ortiz@ug.uchile.cl (N.O.-L.); aracelipleon@gmail.com (A.P.-L.); javiera.favi@ug.uchile.cl (J.F.); larialempy@gmail.com (L.A.); 2Graduate School, Faculty of Medicine, Universidad de Chile, Santiago 8380453, Chile; 3Section of Gastroenterology, Department of Medicine, Hospital Clínico Universidad de Chile, Santiago 8380456, Chile; dannette.vania@uchile.cl (D.G.F.); jponiachik@hcuch.cl (J.P.); 4Department of Pathology, Hospital Clínico Universidad de Chile, Santiago 8380456, Chile; lcarrenotoro@gmail.com; 5Program of Microbiology and Mycology, Institute of Biomedical Sciences, Faculty of Medicine, Universidad de Chile, Santiago 8380453, Chile; alezazueta28@gmail.com (A.Z.); fmagne@uchile.cl (F.M.); 6Medical Technology, Faculty of Medicine Clínica Alemana, Universidad del Desarrollo, Santiago 7630426, Chile; 7Laboratory of Immunogastroenterology II, Department of Clinical Biochemistry and Immunology, Faculty of Pharmacy, Universidad de Concepción, Concepción 4070386, Chile

**Keywords:** alcoholic liver disease, autoimmune hepatitis, fatty liver, fibrosis progression, hepatic inflammation, liver cirrhosis, liver diseases, metabolic dysfunction-associated steatotic liver disease, mast cells, non-alcoholic fatty liver disease

## Abstract

Metabolic dysfunction-associated steatotic liver disease (MASLD) has become highly prevalent worldwide, and its pathogenesis and progression mechanisms remain incompletely understood. An increased activation of innate immune cells in the liver contributes to hepatic fibrogenesis via a chronic loop of inflammation and regeneration processes. Among them are mast cells (MCs), whose role in hepatic cirrhosis secondary to MASLD remains poorly studied. Our aim was to evaluate differences in MC density in cirrhotic liver tissue among patients with MASLD and other chronic liver disease etiologies. For this, a retrospective study of MC count was performed in cirrhotic liver explants obtained from MASLD, alcohol-related liver disease (ALD), and autoimmune hepatitis (AIH). We included a control group of subjects without liver damage. Tryptase-positive MCs were identified by indirect immunofluorescence and quantified as MC density per low-power field (MC/LPF). Group differences were analyzed using the Kruskal–Wallis test with Dunn’s multiple comparisons, considering *p* < 0.05 as statistically significant. A significantly higher MC density was observed in MASLD, ALD, and AIH patients compared with the control group. The group analysis showed that ALD patients exhibited higher MC density than AIH, with no observed difference between ALD and MASLD. MC density was correlated positively with tobacco smoking and alcohol use in the full analyzed group, suggesting them as risk factors of high MC liver infiltration. We conclude that MC density is augmented in MASLD-related cirrhosis, highlighting potential links between lifestyle factors and MC-mediated hepatic inflammation. Future studies should explore the mechanisms driving this association and evaluate whether targeting MCs could help mitigate fibrosis progression.

## 1. Introduction

Metabolic dysfunction-associated steatotic liver disease (MASLD), formerly non-alcoholic fatty liver disease (NAFLD), affects about 25% of the global population and up to 60% of individuals at metabolic risk [1]. It is now one of the leading causes of cirrhosis and hepatocellular carcinoma, with progression from steatosis to advanced fibrosis driven by metabolic and immune-mediated mechanisms [2,3,4]. Other frequent non-infectious etiologies of chronic liver disease include alcohol-related liver disease (ALD) and autoimmune hepatitis (AIH) [5].

Mast cells (MCs) have emerged as important immune cells in liver pathology owing to their pleiotropic roles in inflammation and tissue remodeling [6]. Their fibrogenic activity has been demonstrated in multiple organs, including the liver, where MC-derived mediators drive the activation of fibroblasts and the deposition of extracellular matrix [7,8]. However, their role in MASLD remains poorly characterized [9]. In the liver, MC density increases during chronic injury, and activated MCs release histamine, tryptase, and cytokines that perpetuate inflammation and fibrosis [10,11]. They are mainly located within portal tracts around hepatic vessels and bile ducts, where their recruitment appears to be promoted by inflammatory stimuli [12].

Recent evidence indicates that hepatic MC density is elevated in patients with MASLD, correlating with liver damage progression [13]. However, its role in advanced stages of fibrosis, particularly cirrhosis, remains poorly understood, and no previous study has directly compared hepatic MC density among MASLD and other non-infectious etiologies of liver cirrhosis.

The present study aimed to evaluate and compare the differences in MC density in liver cirrhotic tissue from patients with MASLD and non-infectious etiologies, such as alcohol-related liver disease (ALD), and autoimmune hepatitis (AIH), to explore their potential contribution to the pathophysiology of advanced liver disease.

## 2. Results

### 2.1. Characterization of the Participants

A total of 35 participants were included in this cross-sectional study (age range: 29–73 years; mean age: 38.7 years; 45.7% women and 54.2% men) (Table 1). The cohort comprised 10 patients with MASLD, 10 with ALD, 10 with AIH, and 5 control subjects (CS) with histologically normal livers. There were no statistically significant differences among groups in age, BMI, sex distribution, or MELD score at the time of liver transplantation (*p* = 0.400, *p* = 0.200, *p* = 0.052, and *p* = 0.600, respectively). The MASLD group had a higher Charlson Comorbidity Index (CCI) compared to the other groups, whereas alcohol use was significantly higher in the ALD group (*p* < 0.001).

### 2.2. Differences in MC Density in Cirrhotic Liver Explants from Patients with MASLD, ALD, AIH, and CS

Representative immunofluorescence images of tryptase-positive MCs in CS, MASLD, ALD, and AIH liver tissue are shown in Figure 1. MC density was significantly higher in MASLD (median 36.7 [IQR 29.2–40.1]; *p* = 0.0019), ALD (45.5 [39.7–61.0]; *p* = 0.00028), and AIH (25.0 [23.7–39.1]; *p* = 0.028) compared with controls (18.5 [17.7–18.8]) (Figure 2). ALD exhibited a significantly higher MC density than AIH (*p* = 0.017), whereas the differences between MASLD and ALD (*p* = 0.094) and between MASLD and AIH (*p* = 0.320) were not statistically significant.

### 2.3. Correlation Between MC Density and Clinical Characteristics

MC density showed a positive correlation with smoking (r = 0.437, *p* < 0.01) and with alcohol use (r = 0.449, *p* < 0.01) (Figure 3). No significant correlations were found between MC density and age, BMI, or sex.

## 3. Discussion

This study investigated the role of MCs in liver cirrhosis secondary to MASLD and other non-infectious chronic liver diseases. While previous research has examined MC involvement in metabolic dysfunction-associated steatohepatitis (MASH), their contribution to the stage of cirrhosis has not been explicitly addressed. We found that MC density was significantly increased in cirrhosis secondary to MASLD compared with CS, and similarly elevated in other non-infectious chronic liver diseases. Notably, to our knowledge, this is the first study to directly compare hepatic MC density between patients with cirrhosis secondary to MASLD and those with cirrhosis secondary to ALD. A comparable MC density was observed in both groups, suggesting that its role in common liver inflammatory processes contributes to disease progression.

Our findings are consistent with preclinical evidence in diverse murine models showing that MCs influence liver fibrosis in MASLD. For instance, Kennedy et al. demonstrated that MC depletion in a murine Western diet model prevents microvesicular steatosis and fibrosis development, supporting a pro-fibrogenic role for MCs in this model of liver injury [14]. Similarly, treatment with a chymase inhibitor—one of the proteases produced by liver MCs—prevented the development and progression of steatohepatitis in rats fed a high-fat and high-cholesterol diet [15]. Interestingly, in human studies, Lombardo et al. [13] reported a direct correlation between hepatic MC density and fibrosis stage in MASH, supporting the involvement of MCs in fibrogenesis in this clinical context. In line with this, Lewandowska et al. reported in an autopsy study that mast cells located in portal areas and fibrous septa were implicated in the pathogenesis of liver fibrosis in patients with MASLD without fibrosis or with fibrosis stage F3 or lower [16]. Collectively, these findings indicate that MCs are active contributors to MASLD pathogenesis and represent a promising therapeutic target for halting disease progression.

Despite the increased MC density in AIH compared to CS, the lower density observed in AIH relative to ALD may be attributable to the routine use of immunosuppressive therapy in AIH, particularly glucocorticoids [17]. Glucocorticoids have been shown to reduce tissue MC numbers by downregulating stem cell factor production, a critical factor for MC survival [18]. This mechanism could underlie the relatively lower MC counts observed in AIH compared to MASLD and ALD, despite equivalent disease stages (i.e., cirrhosis). The similar MC density observed in ALD and MASLD suggests that shared inflammatory and fibrogenic pathways may drive MC recruitment in these metabolic and alcohol-related conditions.

We also found a positive correlation between MC density and both tobacco smoking and alcohol use, suggesting that these lifestyle factors may promote MC recruitment to the liver due to their inflammatory response amplification. However, these findings are hypothesis-generating and should be confirmed in larger, prospectively designed studies. While the association between alcohol consumption and hepatic inflammation is well established, the potential role of tobacco in modulating hepatic MCs has not been directly explored. Kaukinen et al. reported increased tryptase-positive MCs in the skin of smokers, suggesting that tobacco exposure may enhance MC proliferation or activation in certain tissues [19]. This underscores the importance of evaluating behavioral interventions targeting addictive behaviors in parallel with pharmacological strategies to mitigate MC-mediated liver injury.

MCs are abundant in the digestive mucosa and are essential components of the intestinal barrier. They regulate epithelial integrity and function, calibrate innate and adaptive mucosal immunity, and sustain the neuroimmune crosstalk that underpins gut function [20,21]. MCs are also increasingly recognized as modulators of the gut–liver axis across liver diseases [22]. This axis comprises bidirectional communication among the gastrointestinal tract, its microbiota, and the liver, and is strongly implicated in MASLD [23]. Future studies should delineate the mechanisms by which MCs influence this axis in MASLD, particularly through interactions with the microbiota and their contributions to disease progression and therapeutic response. These mechanistic considerations are speculative and were not directly evaluated in the present study.

This study offers novel insights into the role of MCs in advanced liver disease across different etiologies, including MASLD, ALD, and AIH. To our knowledge, this is the first work to directly compare hepatic MC density among non-infectious cirrhosis etiologies, thereby addressing an important knowledge gap. The inclusion of control subjects with histologically normal livers strengthens the validity of these findings. Moreover, identifying comparable MC accumulation across these conditions underscores shared inflammatory mechanisms that may inform future diagnostic and therapeutic strategies targeting MC-mediated pathways in advanced liver disease.

Despite our study’s limitations—namely, the small sample size, single-center design, use of end-stage explant tissue, and potential confounding by etiology-specific therapies and differences in sample type (explants versus biopsies)—future studies should include larger multicenter cohorts encompassing different disease stages. Prospective and standardized assessment of alcohol and tobacco exposure, medication use, and cardiometabolic comorbidities, together with automated area-normalized and compartment-specific mast cell quantification, phenotyping, and spatial analyses in longitudinal designs, will be crucial for strengthening causal inference. Additionally, histological factors such as the stage of fibrosis and necroinflammatory activity may also influence MC accumulation, and this aspect should be explored in future studies.

In conclusion, our results support a pathogenic role for MCs in advanced MASLD, with MC density levels comparable to those observed in ALD-related cirrhosis. Further studies with larger, multicenter cohorts and longitudinal designs are needed to confirm these findings, clarify underlying mechanisms, and evaluate the therapeutic potential of targeting MCs—particularly in advanced MASLD—while also assessing the impact of modifying lifestyle factors that may influence MC recruitment and activation in the liver.

## 4. Materials and Methods

### 4.1. Study Design and Setting

A retrospective, single-center observational analysis was performed on tissue explants collected and stored at a large university hospital between 2017 and 2022. The manuscript adheres to the STROBE guidelines for cross-sectional studies.

### 4.2. Participants and Sampling

Cases were identified through the institutional pathology database. Eligible participants were adults (>18 years) of both sexes with cirrhosis of various etiologies (MASLD, ALD, or AIH) who underwent liver explantation. Control subjects were adults who underwent liver biopsy without evidence of liver damage, performed in the context of benign biliary pathology. To ensure balanced comparisons across etiologies, a representative subset of ten cases per group (MASLD, ALD, and AIH) and five control samples was selected from the available pool meeting all inclusion criteria. All diagnoses were confirmed by a pathologist from the Department of Pathology at HCUCH. Patients with acute liver failure or multiple concurrent causes of cirrhosis (e.g., concomitant viral hepatitis, drug-induced liver injury) were excluded.

### 4.3. Preparation and Deparaffinization of Liver Tissue Samples

Formalin-fixed, paraffin-embedded liver tissue sections were deparaffinized and rehydrated following a previous standardized protocol in the Lab. Samples were immersed in xylene twice for 5 min each, followed by a third immersion in xylene for 10 min. They were then rehydrated through a graded ethanol series, with each step lasting 5 min: three washes in 100% ethanol, three in 95% ethanol, and two in 70% ethanol. Finally, the sections were rinsed with distilled water to remove residual ethanol.

### 4.4. Indirect Immunofluorescence for MC Tryptase

Antigen retrieval was performed by incubating the sections in sodium citrate solution (pH 6.0) for 20 min at 100 °C. Samples were then rinsed twice with double-distilled water for 5 min each, and the tissue areas were outlined with a hydrophobic marker. Permeabilization was carried out with 0.3% PBS–Triton for 10 min, followed by blocking with Dako fluorescent mounting medium (Agilent Technologies, Santa Clara, CA, USA) for an additional 10 min. The primary anti-tryptase antibody (1:300; Biotin Anti-Mast Cell Tryptase antibody [EPR9522], ab151757, Abcam, Cambridge, UK) was applied and incubated overnight at 4 °C in a humidified chamber. For negative controls, the primary antibody was replaced with antibody diluent. After incubation, sections were washed three times in 0.3% PBS–Triton for 5 min each, then incubated in the dark at room temperature with a rabbit secondary antibody conjugated to Alexa Fluor 594 (1:5000; Invitrogen, Waltham, MA, USA). Following three additional washes with 0.3% PBS–Triton for 5 min each, nuclei were counterstained with DAPI (1:100; substock 1:500; Sigma Aldrich, London, UK) for 1 min. Finally, sections were rinsed in 1× PBS and mounted with Dako fluorescent mounting medium.

### 4.5. MC Quantification

Immunofluorescence images were acquired using an epifluorescence microscope (Nikon, Tokyo, Japan) with a 10× objective under identical exposure time and gain settings for all samples to ensure comparability. For each liver section, five non-overlapping fields were randomly selected and captured. Tryptase-positive mast cells (MCs) were manually counted using ImageJ software, version 6.0 (National Institutes of Health, Bethesda, MD, USA). by two independent blinded observers. The density of MCs was expressed as the number of mast cells per low-power field (MCs/LPF), following previously published methods in liver histology [24]. For each sample, the mean value obtained from the five fields was used for subsequent statistical analyses.

### 4.6. Statistical Analysis

Group comparisons were performed using the Kruskal–Wallis test, followed by Dunn’s post hoc analysis to identify specific group differences. Correlation analyses were performed using Pearson’s correlation coefficient after confirming the normal distribution of the variables, as this method is appropriate for assessing the strength and direction of linear relationships between continuous variables. All statistical analyses were performed in R software (version 4.1.2; R Foundation for Statistical Computing, Vienna, Austria), with statistical significance set at *p* < 0.05.

## Figures and Tables

**Figure 1 ijms-27-00392-f001:**
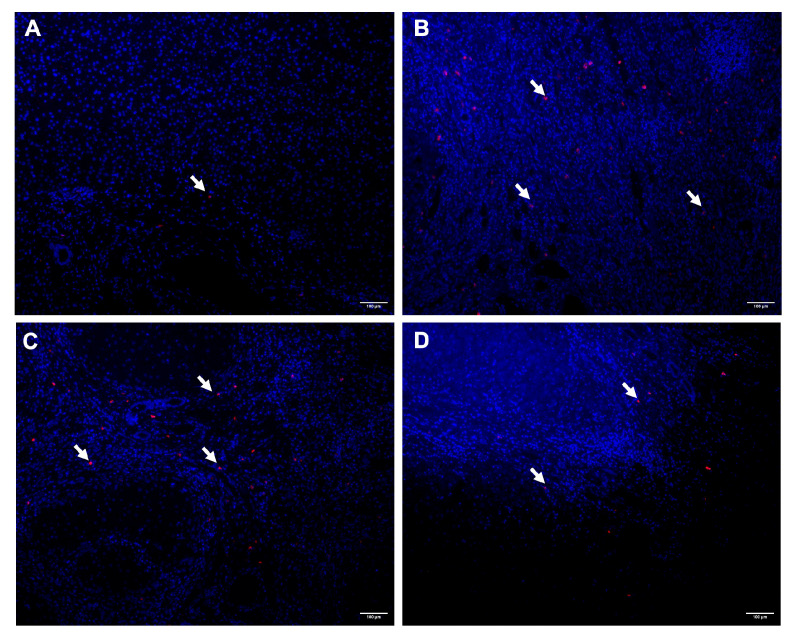
Representative fluorescence micrographs showing tryptase-positive MCs (red) and nuclei stained with DAPI (blue) in liver tissue from: (**A**) CS without liver damage, (**B**) cirrhosis secondary to MASLD, (**C**) cirrhosis secondary to ALD, and (**D**) cirrhosis secondary to AIH. White arrows indicate tryptase-positive MCs. Images were acquired at the same magnification under epifluorescence microscopy. Scale bars: 100 µm. Abbreviations: ALD, alcohol-related liver disease; AIH, autoimmune hepatitis; CS, control subject; MC, mast cell; MASLD, metabolic dysfunction-associated steatotic liver disease.

**Figure 2 ijms-27-00392-f002:**
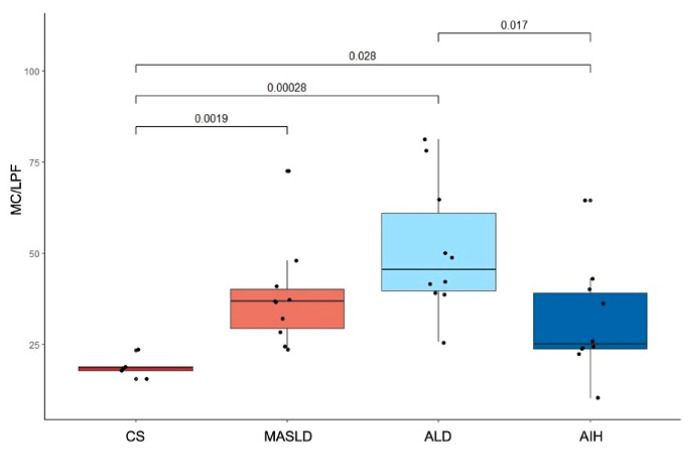
Mast cell density is increased in liver explants from patients with MASLD, ALD, and AIH compared to healthy controls. Box plots depict medians, interquartile ranges, and whiskers (1.5 × IQR); dots represent individual values. Group comparisons were performed using the Kruskal–Wallis test with Dunn’s post hoc test (*p*-values shown), with *p* < 0.05 considered statistically significant. Abbreviations: AIH, autoimmune hepatitis; ALD, alcohol-related liver disease; CS, control subject; MASLD, metabolic dysfunction-associated steatotic liver disease; MC/LPF, mast cells per low-power field.

**Figure 3 ijms-27-00392-f003:**
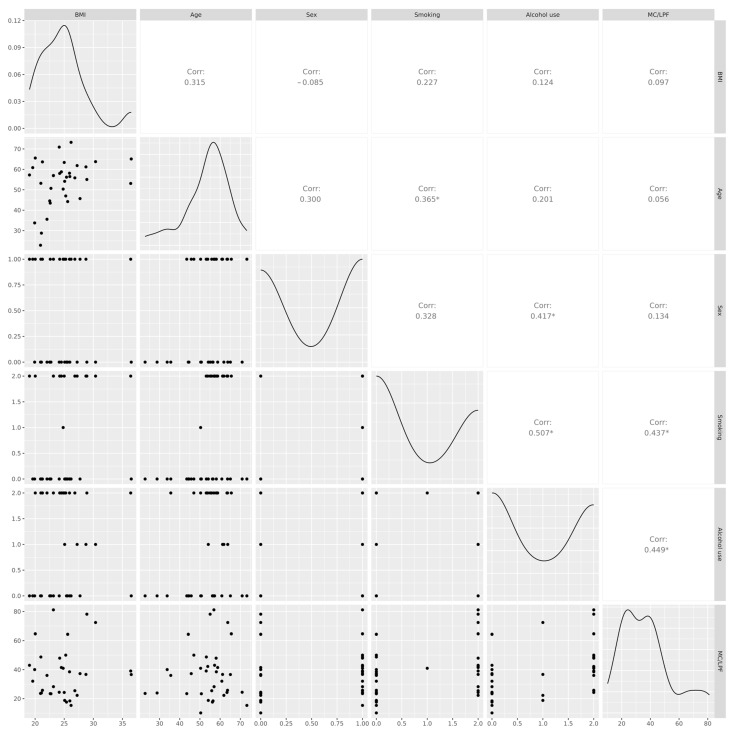
Pairwise relationships between MC density and clinical characteristics. The scatterplot matrix displays BMI, age, sex, smoking, alcohol use, and MC/LPF. The lower-left triangle shows bivariate scatterplots with smoothed trend lines; the diagonal shows kernel density estimates; and the upper-right triangle reports Pearson correlation coefficients. Asterisks denote nominal significance (*p* < 0.05). Abbreviations: BMI, body mass index; MC/LPF, mast cells per low-power field.

**Table 1 ijms-27-00392-t001:** Clinical and demographic characteristics of the study participants.

	CS (*n* = 5)	MASLD (*n* = 10)	ALD (*n* = 10)	AIH (*n* = 10)	*p*-Value
**Age** (mean, IQR)	56.2 (54.1–56.5)	59.4 (50.3–63.8)	55.4 (53.2–58.1)	47.3 (33.8–61.9)	0.400
**Male sex**	1 (20%)	7 (70%)	8 (80%)	3 (30%)	0.052
**BMI**	25.4 (25.1–26.0)	24.5 (23.1–28.7)	25.3 (23.1–26.8)	21.3 (20.9–25.0)	0.200
**Alcohol use**					
Never	4	6	0	6	**<0.001**
Suspended *	1	4	10	4	
Active	0	0	0	0	
**Smoking**					
Never	5	5	3	7	0.070
Suspended *	0	4	7	3	
Active	0	1	0	0	
**Charlson Comorbidity Index** (mean, IQR)	3.0 (1.0–3.0)	5.0 (4.0–6.0)	4.0 (4.0–6.0)	3.5 (3.0–6.0)	**0.010**
**MELD score at liver transplant** (mean, SD)	N/A	23.2 ± 7.82	26.2 ± 5.65	26.4 ± 6.32	0.600

* Suspended: abstinence ≥ 12 months before liver transplantation. Abbreviations: ALD, alcohol-related liver disease; BMI, body mass index; AIH, autoimmune hepatitis; IQR, interquartile range; MASLD, metabolic dysfunction-associated steatotic liver disease; N/A, not applicable; SD, standard deviation.

## Data Availability

The data presented in this study are available on request from the corresponding author. The data are not publicly available due to ethical reasons.

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
