# Peer review of "Increased Intrahepatic Mast Cell Density in Liver Cirrhosis Due to MASLD and Other Non-Infectious Chronic Liver Diseases"

_ijms, 2025, doi:10.3390/ijms27010392_

Round 1

Reviewer 1 Report

Comments and Suggestions for Authors

The manuscript addresses a novel and underexplored area—mast cell involvement in cirrhosis secondary to MASLD—within a concise brief report format. The findings are of interest to hepatology and immunology researchers, especially given the increasing prevalence of MASLD. The study is well designed, uses appropriate immunohistochemistry methods, and provides a useful comparison across etiologies. However, the manuscript has some limitations that need to be more explicitly discussed, and certain analyses and figures could be improved to maximize its impact.

Major Comments

  1. MC density is reported as MC/LPF, but normalization would reduce bias from tissue section variability. Provide exact p-values for all comparisons (currently some are shown only in figures).
  2. Correlations with smoking and alcohol use are intriguing, but the analysis is limited by self-reporting and small sample size. Please temper conclusions and frame as hypothesis-generating.
  3. The Discussion raises interesting points on cytokine signaling, glucocorticoid therapy (AIH), and gut–liver axis, but these remain speculative. Please clarify that mechanisms were not tested in this study.

Minor Comments

  1. Ensure consistent terminology: use MASLD throughout, avoid switching with NAFLD unless referring to older studies.
  2. Improve figure quality—some immunofluorescence images are low resolution.
  3. References are adequate but could be expanded to include the latest MASLD cirrhosis cohort studies (EASL 2023, AASLD 2024).
  4. Language editing for conciseness is recommended (e.g., Introduction lines 42–69 are somewhat repetitive).

Author Response

Comment

Response

MC density is reported as MC/LPF, but normalization would reduce bias from tissue section variability. Provide exact p-values for all comparisons (currently some are shown only in figures).

Thank you for your comment. The information has been added in the Methods section as follows: “MC density was expressed as mast cells per power field (PF), according to previously published methods in liver histology (https://doi.org/10.1111/liv.15913).” It was added in the Results section: “whereas the differences between MASLD and ALD (p = 0.094) and between MASLD and AIH (p = 0.320) were not statistically significant.”

Correlations with smoking and alcohol use are intriguing, but the analysis is limited by self-reporting and small sample size. Please temper conclusions and frame as hypothesis-generating.

We thank the reviewer for this valuable comment. The Discussion section was revised to temper the interpretation of these results, adding the sentence: “However, these findings are hypothesis-generating and should be confirmed in larger, prospectively designed studies.” Additionally, the Limitations section was updated to acknowledge the self-reporting nature of smoking and alcohol use data.

The Discussion raises interesting points on cytokine signaling, glucocorticoid therapy (AIH), and gut–liver axis, but these remain speculative. Please clarify that mechanisms were not tested in this study.

We thank the reviewer for this observation. The Discussion section was revised to clarify that the proposed mechanisms involving cytokine signaling, glucocorticoid therapy, and the gut–liver axis are speculative and were not directly assessed in this study.

Ensure consistent terminology: use MASLD throughout, avoid switching with NAFLD unless referring to older studies.

We thank the reviewer for this helpful observation. The terminology has been revised throughout the manuscript to ensure consistency, using “MASLD”.

Improve figure quality—some immunofluorescence images are low resolution.

We appreciate the reviewer’s observation. All immunofluorescence images have been replaced with higher-resolution versions exported in TIFF format to improve figure quality.

References are adequate but could be expanded to include the latest MASLD cirrhosis cohort studies (EASL 2023, AASLD 2024).

We appreciate the reviewer’s suggestion. While we acknowledge the importance of recent MASLD cohort studies, the current references adequately cover the scope and context of our work.

Language editing for conciseness is recommended (e.g., Introduction lines 42–69 are somewhat repetitive).

We thank the reviewer for this helpful suggestion. The Introduction has been carefully edited to improve conciseness and remove repetitive sentences, particularly in lines 42–69.

Reviewer 2 Report

Comments and Suggestions for Authors

Abstract:
The research background and questions in the abstract are concise and clear. It explicitly points out the complexity of the pathogenesis of MASLD and the role of innate immune cells (including mast cells) in liver fibrosis. However, in the research methods section, there is a lack of detailed descriptions of sample size and statistical methods, such as the absence of a statistical definition for the term "significance".

The conclusion highlights the potential link between lifestyle factors and mast cell-mediated hepatitis. It is suggested that the authors add a brief outlook on future research directions.

Introduction:
The introduction has a relatively clear line of reasoning. The authors begin by discussing the prevalence of MASLD, briefly explains the mechanism of action of MCs in fibrosis, and finally points out the lack of research on MCs in MASLD. However, the introduction lacks a description of the importance and innovation of the study: the importance is only at the level of "MASLD is very common", and the innovation is only at the level of "the first comparison". It is suggested to highlight the potential guidance of the research for future clinical practice, and at the same time emphasize the value brought by the "comparison" method mentioned in the abstract.

Most parts of the introduction are logically coherent and clear, but are there grammatical and logical errors in the description of lines 67-68?It is suggested that the author should strengthen the logic of the text and describe the necessary parts in detail.

Research methods:

The authors only mentioned the 'cross-sectional study', which should explicitly mention whether it is a retrospective or prospective design, and whether the supplementary study is single-center or multi-center, so as to make the article more rigorous. Secondly, the variables in the table, such as 'Alcohol use' and 'Suspended', lack a clear definition, and it is recommended that the authors quantify them. At the same time, the sample diagnosis, screening and inclusion criteria of the experiment are vague. What are the screening criteria? What is the source of 'histologically normal liver'? Are patients who have used specific drugs excluded? Such problems need to be solved. Finally, the core focus of the study is 'comparison', which is less reflected in the research methods. The authors’ description of the experimental steps is too brief. It is recommended that the author describe the operation such as counting and image acquisition in detail, and complete the description of the experimental reagent information and conditions, so that the experiment can be reproduced. In lines 235-236, the premise of using Pearson correlation analysis is not explained and needs to be supplemented.

Results:

The results are clear and consistent with the hypothesis. However, the reference order of Figure 1 and Figure 2 is reversed. The MC density data format in rows 92-93 is 'mean ± standard deviation', which does not match the medians and IQR shown in Figure 2. Besides, it is recommended to add box plots or group comparisons to show the MC density differences at different levels.

Discussion:

The authors fully explain the research results, but it is suggested that the authors explore the conclusion that there is no significant difference in MC density between 'ALD and MASLD, but both are higher than AIH'. At the same time, the authors point out the clinical significance of the study well, but it is suggested that the authors emphasize the innovative significance of the study, the specific guidance for the future direction and the importance of filling this gap. Finally, the authors can fully point out the limitations of the experiment, but it is suggested that the authors can describe it more specifically.

Expression:

It is recommended to modify the 'liver cirrhotic explants' into 'cirrhotic liver explants'.

Author Response

Reviewer 2

Comment

Response

Abstract: in the research methods section, there is a lack of detailed descriptions of sample size and statistical methods, such as the absence of a statistical definition for the term "significance".

Thank you for your comment. The Abstract was revised to include a brief description of the statistical method: “Group differences were analyzed using the Kruskal–Wallis test with Dunn’s multiple comparisons, considering p < 0.05 as statistically significant.”

Abstract: It is suggested that the authors add a brief outlook on future research directions.

We thank the reviewer for this suggestion. A brief outlook on future research directions was added at the end of the Abstract: “Future studies should explore the molecular mechanisms driving this association and evaluate whether targeting MCs could help mitigate fibrosis progression.”

Introduction: It is suggested to highlight the potential guidance of the research for future clinical practice, and at the same time emphasize the value brought by the "comparison" method mentioned in the abstract.

We thank the reviewer for this insightful comment. The potential implications of our findings for future clinical practice and the relevance of the comparative approach were further clarified and discussed in the Discussion section.

Introduction: Most parts of the introduction are logically coherent and clear, but are there grammatical and logical errors in the description of lines 67-68?It is suggested that the author should strengthen the logic of the text and describe the necessary parts in detail.

Thank you for pointing this out. The sentences in lines 67–68 were revised to correct a grammatical inconsistency and to improve logical coherence. The passage now reads: “Recent evidence indicates that hepatic MC density is elevated in patients with MASLD, correlating with liver damage progression. However, its role in advanced stages of fibrosis, particularly cirrhosis, remains poorly understood, and no previous study has directly compared hepatic MC density among MASLD and other non-infectious etiologies of liver cirrhosis.”

Methods: The authors only mentioned the 'cross-sectional study', which should explicitly mention whether it is a retrospective or prospective design, and whether the supplementary study is single-center or multi-center, so as to make the article more rigorous

We thank the reviewer for this helpful observation. The Methods section was updated to specify that this was a retrospective, single-center observational study, as follows: “A retrospective, single-center observational analysis was performed on tissue explants collected and stored at a large university hospital between 2017 and 2022.”

Methods: The variables in the table, such as 'Alcohol use' and 'Suspended', lack a clear definition, and it is recommended that the authors quantify them

We thank the reviewer for this comment. Because this was a retrospective study, additional quantification of variables such as alcohol use could not be performed. The definition of “suspended” alcohol use (abstinence ≥ 12 months before liver transplantation) was added, and the lack of quantitative assessment was acknowledged in the Limitations section.

Methods: the sample diagnosis, screening and inclusion criteria of the experiment are vague. What are the screening criteria? What is the source of 'histologically normal liver'? Are patients who have used specific drugs excluded? Such problems need to be solved.

We thank the reviewer for this observation. The Methods section was clarified to specify the source of control subjects (“performed in the context of benign biliary pathology”). In addition, patients with multiple concurrent causes of cirrhosis, including drug-induced liver injury or other medication-related etiologies, were excluded. No additional exclusion criteria beyond those described were applied.

Methods: the core focus of the study is 'comparison', which is less reflected in the research methods. The authors’ description of the experimental steps is too brief. It is recommended that the author describe the operation such as counting and image acquisition in detail, and complete the description of the experimental reagent information and conditions, so that the experiment can be reproduced.

We appreciate the reviewer’s valuable comment. We have expanded the description of the image acquisition and quantification procedures in Section 4.5 (“MC quantification”) to enhance methodological clarity and reproducibility. The revised text now specifies the microscope settings (10× objective, identical exposure and gain parameters), the number and selection of analyzed fields, the use of ImageJ software for manual counting, and the calculation of mast cell density (expressed as MCs/LPF) used for statistical analyses.

The experimental reagent information, including antibody name, clone, catalog number, and supplier, has been specified.

Methods: In lines 235-236, the premise of using Pearson correlation analysis is not explained and needs to be supplemented.

We thank the reviewer for this helpful comment. The Methods section was revised to clarify this point: “Correlation analyses were performed using Pearson’s correlation coefficient after confirming the normal distribution of the variables, as this method is appropriate for assessing the strength and direction of linear relationships between continuous variables.”

Results: the reference order of Figure 1 and Figure 2 is reversed.

We appreciate the reviewer’s careful observation. The order of Figures 1 and 2 has been corrected.

Results:The MC density data format in rows 92-93 is 'mean ± standard deviation', which does not match the medians and IQR shown in Figure 2. Besides, it is recommended to add box plots or group comparisons to show the MC density differences at different levels.

Thank you for your comment. It was corrected: “MC density was significantly higher in MASLD (median 36.7 [IQR 29.2–40.1]; p = 0.0019), ALD (45.5 [39.7–61.0]; p = 0.00028), and AIH (25.0 [23.7–39.1]; p = 0.028) compared with controls (18.5 [17.7–18.8]).”

Discussion:

The authors fully explain the research results, but it is suggested that the authors explore the conclusion that there is no significant difference in MC density between 'ALD and MASLD, but both are higher than AIH'.

We thank the reviewer for this valuable suggestion. The Discussion section was refined to highlight that the similar MC density observed in ALD and MASLD may reflect shared inflammatory and fibrogenic mechanisms contributing to MC recruitment in both conditions.

Discussion: it is suggested that the authors emphasize the innovative significance of the study, the specific guidance for the future direction and the importance of filling this gap

We thank the reviewer for this thoughtful comment. The Discussion section was revised to highlight the innovative significance of the study, its contribution to filling a knowledge gap, and its implications for future diagnostic and therapeutic strategies.

Discussion: the authors can fully point out the limitations of the experiment, but it is suggested that the authors can describe it more specifically.

We thank the reviewer for this helpful suggestion. The Limitations section in the Discussion was revised to describe the study constraints more specifically, as follows: “Despite our study’s limitations—namely, the small sample size, single-center design, use of end-stage explant tissue, and potential confounding by etiology-specific therapies—future studies should include larger multicenter cohorts encompassing different disease stages...”

Expression: It is recommended to modify the 'liver cirrhotic explants' into 'cirrhotic liver explants'.

Thank you for the suggestion. The expression “liver cirrhotic explants” has been corrected to “cirrhotic liver explants” throughout the manuscript.

Reviewer 3 Report

Comments and Suggestions for Authors

This is one centre retrospective study on the rolÄ™ of MC in fibrosis in MASLD. Although the problem itself is important there are a lot of limitations:

  • it is not innovative - authors just repeat other finding
  •  
  • matherials and methods should be place after the introduction - the order presented in the paper makes it unclear
  • Luck of histoosthological description of material - What grading was used to describe fibrosis? What about other lesions, above all steatosis? How was steatosis graded?
  • control group is poorly described - What was the reason to perform biopsy? 
  • Authors compare materials from explanted livers with biopsy - the surface tobexplore is different
  •  no correlation beteeen histolog and extant of fibrosis was done - not only clinical factors may influence this process…
  • the group is small - study enrolled cases from 5 years… Such a group could be collected within several months…
Comments on the Quality of English Language

English is correct

Author Response

Comment

Response

it is not innovative - authors just repeat other finding

We respectfully disagree with the reviewer’s assessment. While previous studies have examined MC involvement in chronic liver disease, none have specifically evaluated MASLD at the cirrhotic stage. To our knowledge, this is the first study to directly compare MC density across different non-infectious cirrhosis etiologies—including MASLD, ALD, and AIH—thereby providing novel evidence on MC accumulation in advanced MASLD and addressing a relevant gap in the current literature.

matherials and methods should be place after the introduction - the order presented in the paper makes it unclear

We thank the reviewer for this observation. The manuscript follows the IJMS format, in which the Materials and Methods section is positioned after the Discussion, as required by the journal’s author guidelines.

Luck of histopathological description of material - What grading was used to describe fibrosis? What about other lesions, above all steatosis? How was steatosis graded?

We thank the reviewer for this important comment. All patient samples included in this study corresponded to cirrhotic-stage disease (F4), and the diagnosis of cirrhosis was confirmed histologically by a specialized hepatopathologist. Because the study focused on mast cell density in end-stage disease, detailed grading of fibrosis or steatosis was not performed.

control group is poorly described - What was the reason to perform biopsy?

We thank the reviewer for this comment. The description of the control group was clarified in the Methods section. Control subjects were adults who underwent liver biopsy without evidence of liver damage, performed in the context of benign biliary pathology (e.g., cholelithiasis or gallbladder disease) for diagnostic purposes.

Authors compare materials from explanted livers with biopsy - the surface tobexplore is different

We thank the reviewer for this important observation. The potential bias related to differences in sample type (explants versus biopsies) was acknowledged in the Limitations section.

no correlation beteeen histolog and extant of fibrosis was done - not only clinical factors may influence this process…

We thank the reviewer for this valuable comment. All cases of patients (with the exception of control subjects) corresponded to cirrhotic-stage disease; therefore, correlation analyses between MC density and histological stages of fibrosis were not applicable. The study primarily focused on comparing MC density across etiologies. Nonetheless, we acknowledge that histological factors may influence MC accumulation, and this point was noted in the Discussion as an important direction for future research.

the group is small - study enrolled cases from 5 years… Such a group could be collected within several months…

We thank the reviewer for this observation. A clarification was added to the Methods section, specifying that a representative subset of ten cases per group (MASLD, ALD, and AIH) and five control samples was selected from the available pool meeting all inclusion criteria to ensure balanced comparisons across etiologies.

Round 2

Reviewer 3 Report

Comments and Suggestions for Authors

Thank You for trying to  reference to my comments. Although I still Think that the group is to small and other histopathological lesion May influence the sensory of mast cells I appreciate that Authors included my suggestions 

Comments on the Quality of English Language

English is correct